# Influence of Magnesium Oxide on the Structure and Catalytic Activity of the Wustite Catalyst for Ammonia Synthesis

**DOI:** 10.3390/ma15238309

**Published:** 2022-11-23

**Authors:** Artur Jurkowski, Aleksander Albrecht, Dariusz Moszyński, Rafał Pelka, Zofia Lendzion-Bieluń

**Affiliations:** Department of Inorganic Chemical Technology and Environment Engineering, Faculty of Chemical Technology and Engineering, West Pomeranian University of Technology in Szczecin, Piastów Ave. 42, 71-065 Szczecin, Poland

**Keywords:** ammonia synthesis, wustite, magnesium oxide

## Abstract

The influence of a magnesium oxide admixture on the activation process and catalytic activity of the iron catalyst with a wustite structure was investigated during the ammonia synthesis reaction. The incorporation of magnesium oxide into wustite grains is considered to be a structure-forming and activating promoter. It stabilizes the α-Fe structure and increases the activity of the catalysts in the ammonia synthesis reaction. Moreover, magnesium oxide forms a solid solution with the wustite, which slows down the reduction of a catalyst precursor. Similar to calcium and potassium compounds, magnesium oxide is present on the α-Fe surface of the active form of the catalyst. The optimum MgO concentration in the catalyst structure was determined to be 1.2% wt.

## 1. Introduction

High-pressure ammonia synthesis utilizing an iron catalyst was developed over 100 years ago in Germany by Haber and Bosch [1]. This technology stimulated the development of the nitrogen industry and significantly influenced the research on heterogeneous catalysis [2]. To this day, the iron catalyst obtained as a result of magnetite reduction is the best-known catalyst in the world [3]. Current trends in the development of catalytic ammonia synthesis encompass the catalysts obtained from magnetite [4], wustite [5,6], hematite [7], ruthenium [8,9], cobalt [10], and cobalt molybdenum nitrides [11,12].

Research has been carried out on alternative methods of ammonia synthesis, which enable the process to be conducted under mild conditions, e.g., electrochemical or photocatalytic ammonia synthesis [13].

Wustite is iron (II) oxide with the general formula Fe_1−x_O. The non-stoichiometric formula is an effect of the partial oxidation of Fe^2+^ ions into Fe^3+^ ions. Wustite was first used as a precursor for iron catalysts for ammonia synthesis in 1986 by Liu Huazhang [1]. In comparison to the catalysts obtained from magnetite, the iron catalysts obtained by the reduction of wustite are more active and stable in ammonia synthesis and have higher mechanical strength and resistance to poisoning by the impurities in the reactant gas [14]. The type of iron oxide phase present in the catalyst precursor is a key factor influencing the composition and morphology of the reduced form of the catalyst [15].

Outstanding catalytic properties of the wustite-based catalyst are without any doubt associated with its unique structure and the role of promoters. The role of Al_2_O_3_ in the magnetite precursor of the ammonia synthesis catalyst is well known. Alumina is a structural promoter influencing the stability of the α-Fe crystallites obtained after the reduction of magnetite [3,16,17]. It forms a thin layer of Al_2_O_3_ and FeAl_2_O_4_ [18] on the surface of the iron crystallites, which protects them from sintering during ammonia synthesis [19]. Alkaline earth metals can form a spinel structure with magnetite, e.g., calcium ferrite [14] or magnesium ferrite [16]. However, the structure of the wustite precursor does not facilitate a homogeneous distribution of alumina [20]. It diminishes the role of this compound as a structural precursor in this system. Nevertheless, in the literature [3], there are claims that alumina has a significant influence on the active surface restructuring of the wustite-based catalyst [3]. The AlFe_2_O_4_ formed during reduction increases the exposure of the most active iron faces (111) and (211) under the conditions of the ammonia synthesis process.

It was proven that oxides such as CaO, MgO, and ZrO_2_ can be applied as structural promoters in the wustite catalysts [3]. Calcium oxide easily builds into the wustite structure, which also has an impact on its uniform distribution in the reduced form of the catalyst. Due to their small radius, magnesium ions can be built into the structure of wustite [21,22]. Due to the high solubility in the wustite crystal, it should be considered a structural promotor that increases the resistance to sintering and increases the activity. It can be considered that MgO compensates for the role of Al_2_O_3_ as the structural promoter in a wustite-based catalyst. A low concentration of magnesium oxide significantly slows down the reduction of wustite [23]. However, these mechanisms are unknown. There is also no information on how the amount of MgO impacts the reduction process and activity in the ammonia synthesis reaction of the wustite catalyst.

In the present study, magnesium oxide is applied as a promoter in wustite-based iron catalysts for the ammonia synthesis reaction. A set of MgO-promoted catalysts are characterized by X-ray powder diffraction (XRPD), the temperature-programmed reduction method (TPR), inductively coupled plasma optical emission spectrometry (ICP-OES), and X-ray photoelectron spectroscopy (XPS), and their activity is tested within the ammonia synthesis reaction. The influence of the magnesium oxide on the reduction process and the activity of the wustite-based catalyst for the ammonia synthesis reaction are discussed.

## 2. Experimental

### 2.1. Preparation of Catalyst Precursors

The wustite-based precursors of the iron catalyst for ammonia synthesis were prepared by fusion in a laboratory installation. The installation has been described in previous papers [22]. As substrates, we used magnetite, aluminum, magnesium and calcium oxides, potassium nitrate (V), and metallic iron as a reducer of iron (III) ions in magnetite. The fusion was performed with an electric current of 2000 A passing through the mixed substrates. The process duration was 50 min. The molten precursors were poured out into a water-cooled form. The solid product was crushed and sifted to separate the grain fraction with a diameter between 1.0 and 1.2 mm.

### 2.2. Characterization of Catalysts

The XRPD measurements were analyzed in two states: as catalyst precursors (the unreduced form) and after reduction in a hydrogen atmosphere. The phase composition of the prepared precursors was investigated using an X-ray diffractometer (Empyrean, Malvern Panalytical, Malvern, United Kingdom) with a Cu Kα x-ray source (λ_α1_ = 0.1540598 nm, λ_α2_ = 0.1544426 nm). Reduction was carried out in an XRK 900 reaction chamber (Antor Paar, Graz, Austria), which is part of the X-ray diffractometer (X’Pert Pro Philips, Amsterdam, Netherlands) with a Co Kα x-ray source (λ_α1_ = 0.179907 nm, λ_α2_ = 0.179285 nm). Process reduction was performed for 2 h in a 50 cm^3^/min hydrogen flow at a temperature of 500 °C. Scans were taken at room temperature in the scattering 2θ range of 40–120° with a 0.02 step. Phase composition was determined using Panalytical X’Pert HighScore Plus v3.0 software with the ICDD PDF4+ database. Crystallite sizes were calculated based on the Rietveld refinement method.

The molar ratio of the Fe^2+^/Fe^3+^ ions in the precursors was determined by the change in the lattice parameters [24].

The chemical composition of the precursors was examined using the ICP-OES method with a (Perkin Elmer Avio 500, Waltham, MA, USA) spectrometer. An amount of 0.1 g of catalyst precursors was dissolved in 8 mL of hydrochloric acid (35–38% Emprove). This process was performed in a Titan MPS 8 microwave oven with Teflon vessels at 180 °C for 30 min.

The surface compositions of two selected catalysts, W-12 and TA-22, were analyzed with X-ray photoelectron spectroscopy (XPS). The analysis was performed for as-prepared samples as well as for the samples after reduction in a hydrogen atmosphere. The reduction was conducted in a high-pressure cell (HPC) of an ultra-high vacuum (UHV) system. A tablet, 10 mm in diameter, was introduced into the HPC. Hydrogen (99.999 vol.%) was passed through the volume of the reactor at a constant flow of 20 cm^3^/min. The sample was heated to 550 °C for 5 h. After reduction, the sample was transferred under UHV to the analysis chamber of the electron spectrometer. The photoelectron measurements were conducted with Al *K_a_* (hν = 1486.6 eV) radiation in a Prevac system equipped with a Scienta SES 2002 electron energy analyzer operating at constant transmission energy (*E_p_* = 50 eV). The pressure in the analysis chamber was kept under 1∙10^−9^ mbar.

The temperature-programmed reduction process (TPR) was carried out using AutoChem II 2920 (Micromeritics Norcross, GA, USA) equipped with a thermal conductivity detector (TCD) in a quartz reactor using 0.08 g of precursor. Precursors were reduced with a hydrogen/argon mixture (10% vol of hydrogen) with a flow rate of 70 cm^3^/min and a heating rate of 10 °C/min.

The distribution of promoters in the precursors was determined by selective etching using hydrochloric acid [25]. A sample of 0.5 g of the precursor was dissolved in 50 cm^3^ of an aqueous solution of hydrochloric acid. The acid concentration ranged from 0.9% to 36% by mass. The precursor with the hydrochloric acid solution was placed in a shaker in a 200 cm^3^ Erlenmeyer flask. Variable dissolution times and concentrations of hydrochloric acid were applied to obtain different degrees of etching. The solution was filtered, and the concentrations of iron, aluminum, calcium, potassium, and magnesium were evaluated using the ICP-OES method. The concentrations of the promoters inside the wustite grains and in the intergranular spacers were calculated.

The activity of catalysts in the ammonia synthesis was tested in the six-channel reactor [20]. A stoichiometric mixture of nitrogen and hydrogen was obtained by the decomposition of ammonia on a series of three reactors filled with nickel on activated carbon; the decomposition was carried out at 740 °C. The reduction of the precursors was carried out according to the appropriate temperature program. The activity of the catalysts was measured at a temperature of 450 °C under the pressure of 100 bar. Thermostability was evaluated by overheating the catalyst in the atmosphere of the synthesis gas at a pressure of 0.1 MPa over a period of 17 h at a temperature of 600 °C. After completion of the overheating, the determination of the catalyst activities was performed again. The concentration of ammonia in outlet gases was measured with a Siemens Ultramat 6 NDIR analyzer. The rate constant calculated from the Temkin–Pyzhew equation was accepted as a measure of the activity.

## 3. Results and Discussion

The chemical composition and the molar ratio of Fe^2+^/Fe^3+^, denoted as R, are presented in Table 1. The precursors differ mainly in their concentrations of magnesium oxide, which range from 0.00% wt. to 1.54% wt., as well as in their ranges of R, which vary from 5.2 to 9.5. In Table 1, the industrial wustite catalyst is denoted as IND. In precursors with R-values greater than 3.15, the Fe^3+^ ions do not form a separate magnetite phase but are dissolved in the non-stoichiometric Fe_1−x_O phase, which is consistent with the results presented in the literature [23].

The powder X-ray diffraction pattern of the catalyst precursors obtained is presented in Figure 1. The occurrence of the reflex confirms the presence of the wustite phase (ICDD No. 04-003-7164).

The distributions of the promotors in the precursors were determined with the selective etching method using hydrochloric acid [25]. The relationship between the degree of promotor etching and the degree of iron etching evaluated in the precursor W-22 is presented in Figure 2. Results for other precursors are shown in Appendix A. The degree of etching observed for magnesium is almost identical to the degree of iron etching. Therefore, it is concluded that magnesium ions are incorporated into the lattice of the wustite phase. Analysis of the corresponding relations for calcium and aluminum ions suggests that about 50% of these elements are incorporated into the lattice of the wustite phase and the other part of them fills the intergranular spaces, supposedly forming the glassy phase. Potassium is located in the intergranular spaces, which is in line with a former report [24].

Based on these results, the concentrations of promotors in the wustite lattice were calculated (see Table 2). There is a good correlation between the magnesium concentration in the precursor lattice and the total concentration of magnesium oxide in the precursors. The common valence of the iron ions present in the wustite phase and the magnesium ions (namely Fe^2+^ and Mg^2+^) facilitates the substitution of iron by magnesium in the considered phase. The presence of other promotors or R-value has no visible influence on the distribution of magnesium in the precursors.

In the case of aluminum ions, we observe the dependence of the incorporated aluminum ions on the R of the precursors. Precursors with a lower R have a higher concentration of Fe^3+^, which can be easily replaced by ions with a similar valence structure and an ion radius similar to that of aluminum ions. On the other hand, precursors with a higher R have a higher concentration of Fe^2+^, which can be replaced by calcium ions or magnesium. 

The temperature-programmed reduction (TPR) profiles for selected precursors are presented in Figure 3. Although only one peak occurs in each profile, the peak maxima vary. The lowest maximum of 736 °C is observed for the sample W-12, which was formed without a magnesium oxide addition. The higher the magnesium oxide concentration in the precursor, the higher the temperature of the TPR profile maximum. Based on the chemical composition of the catalysts, it can be concluded that the content of MgO significantly affects the course of the reduction process. From the slope of the curves, it can be concluded that the slowest reduction process was observed for the TA-22 catalyst with the highest concentration of magnesium oxide of about 1.5% wt. The TPR profiles are asymmetric for all precursor catalysts since different reduction steps overlap. The reduction degree of each precursor was calculated based on the hydrogen consumption during the TPR processes, and this is presented in Table 3. The reduction degree decreases with an increasing concentration of magnesium oxide in the precursors. This may be a result of the formation of new magnesium and iron phases or magnesium ions decreasing the kinetic parameters of the reduction reaction.

The in situ X-ray diffraction patterns of the catalysts obtained after reduction in a hydrogen atmosphere at 500 °C are shown in Figure 4. At the presented diffractograms, we can distinguish the reflexes originating from four iron crystallographic planes: (110) at 51.9°, (200) at 76.6°, (211) at 98.7°, and (220) at 122.4°. Additionally, in the range between 48 and 50.5°, low-intensity reflexes can be identified. These are attributed to the non-reduced Fe_1−x_O phase. Their intensity increases with the increasing content of magnesium oxide in the catalyst. When comparing diffractograms, an offset is also observed. The shift of the maximum reflex towards higher angular positions for the TA-22 catalyst with the MgO content is about 0.2° in relation to the W-12 catalyst without oxide magnesium (Figure 5). It can, therefore, be concluded that this reflex corresponds to the phase constituting the solid solution of MgFe_1−x_O. Taking into account the size of the Mg^2+^ (0.57 Å) ions in relation to the Fe^2+^ (0.61 Å) ions, we can explain the observed shift. All diffractograms were made under the same conditions. The average sizes of the iron crystallites calculated on the basis of the diffraction data are presented in Figure 6. A substantial increase in this parameter is observed with an increase in the concentration of magnesium oxide in the precursors.

The surface compositions of two selected catalysts, varying substantially in their concentrations of magnesium, namely W-12 and TA-22, were studied with X-ray photoelectron spectroscopy. These materials were examined before and after their reduction in a hydrogen atmosphere. The survey spectra acquired before and after the reduction process are compared in Figure 7 for both catalysts.

In the catalysts examined before reduction, the analysis of the surface composition revealed the presence of oxygen and iron atoms. Apart from these, the existence of potassium and carbon atoms and small concentrations of calcium, aluminum, and silicon atoms were also identified on the surface. These results are in general agreement with the chemical composition studies of the catalysts performed with the OES-ICP method, wherein the presence of promoter phases, such as Al_2_O_3_, K_2_O, and CaO, was indicated. In addition, the XPS study showed the presence of an insignificant content of silicon and carbon atoms, which were not determined by the chemical method. It should be noted that in neither of the two tested samples were magnesium atoms found on the surface before the reduction process. In Table 4, the estimated quantitative information about the surface compositions of these catalysts is shown. The atomic concentrations of a given promoter (n_X_) were related to the atomic concentration of potassium (n_K_). The magnesium surface concentrations were estimated based on the intensity of the magnesium Auger peak, Mg KLL, since both magnesium photoelectron peaks, Mg 2s and Mg 2p, overlap substantially with the iron peaks Fe 3s and Fe 3p, respectively. These ratios, calculated for both samples before reduction, are identical, which proves that the surface compositions of both catalysts before the activation process are virtually identical.

The samples W-12 and TA-22 were then subjected to a reduction process in an atmosphere of pure hydrogen at 550 °C. After this process, the surface analysis with X-ray photoelectron spectroscopy was repeated. The acquired spectra demonstrate a significant change in the surface composition after the process. Most notably, the positions and shapes of the iron spectral lines changed. The XPS Fe 2p line shifted by about 3 eV towards lower binding energies, indicating a reduction in the iron ions present in the oxide phases before the process to metallic iron. The reduction while in the hydrogen atmosphere also resulted in the complete removal of carbon atoms from the surface of the catalysts. A characteristic increase in the intensities of the XPS lines originating from calcium atoms is also observed, indicating a significant enrichment of the catalyst’s surface with this element caused by the reduction process. The elements observed on the surface before reduction, such as potassium, aluminum, and silicon, remain on the surface after the reduction process as well.

The most characteristic change, as well as the feature distinguishing the composition of the surface of the W-12 catalyst from the TA-22 catalyst, is the fact that, after the reduction on the surface of the latter catalyst, the presence of magnesium atoms is detected. There is an overlap of the characteristic photoelectron lines coming from magnesium and from iron. Therefore, the spectral line coming from the Auger electrons Mg KLL, which is observed at a binding energy of about 305 eV, was used to identify the presence of magnesium. A very intense peak is observed in the spectrum of the TA-22 catalyst at this binding energy. In the case of the W-12 catalyst, only a very low-intensity line is observed at the same binding energy.

Due to the relatively low-sensitivity coefficients of the magnesium photoelectrons, this element was not identified in the TA-22 catalyst prior to reduction. It is suggested that in the catalyst precursor, magnesium atoms are dispersed in the grain volume. However, after the reduction process, some of these atoms diffuse and deposit on the surface of the catalyst grains.

On the surface of the reduced catalysts, the atomic concentrations of aluminum and silicon related to the atomic concentration of potassium are higher than in the catalysts’ precursors. A substantial enrichment in calcium atoms is also observed in the precursors. It is notably higher for the TA-22 catalyst.

Figure 8 shows a correlation of the MgO concentrations in the precursors with the activity of the catalysts examined at 450 °C under 10 MPa. Two series of catalysts were examined: the data are presented as a black-point display activity for the catalysts reduced at 500 °C, while the red points represent the activity of the catalysts after a thermostability test. The reduction at 600 °C was used to simulate overheating of the catalysts in the reactor. The addition of magnesium oxide had a huge positive impact on the iron-based catalysts for the ammonia synthesis reaction. Each catalyst doped with magnesium had higher activity than the reference catalysts. The highest improvement in activity can be observed for the catalysts with an addition of magnesium oxide in the range from 0.9% to 1.2% by mass. For these catalysts, we can observe an increase in activity of over 47% compared to the reference catalyst. The further increase in the concentration of this promotor caused a decrease in the activity of the obtained catalysts. It is also important that the decrease in the activity of the catalysts after overheating is much smaller for the catalysts promoted with magnesium oxide than the catalysts without this addition. Magnesium oxide, in the wustite catalyst, takes on the role of a structural promoter. A decrease in the activity of the catalysts with MgO content above 1.2% by mass can be associated with a significant increase in the iron crystallites in these catalysts. The specific surface areas of the reduced catalysts at 500 °C were investigated with the method of single-point nitrogen adsorption at the liquid nitrogen temperature. The results varied in a range from 7.1 to 8.6 m^2^/g. The size of the specific surface of the catalyst is influenced not only by the MgO content but also by the content of other promoters in the catalyst grain. The content of these promoters in the catalyst grain depends on the R-value. Catalysts with various R-values were studied, so a direct relationship between the MgO content and the specific surface area is hard to find.

Iron crystallites are formed during the reduction stage. Folke and co-workers [26] suggest that the mechanism for wustite reduction depends on the compactness of the precursors. The promoters stabilize the metastable wustite phase and inhibit thermal disproportion. On the basis of the presented results of the reduction of magnesium oxide-promoted wustite catalysts, carried out in situ in a reaction chamber combined with the XRD diffractogram, it can be concluded that magnesium oxide forms a solid solution with wustite, which is reduced significantly more slowly than catalysts without magnesium. Reflections belonging to the magnetite phase are also not observed, which may suggest that the process of reducing wustite proceeds directly to iron, and there is no thermal disproportion reaction. However, the TPR curves have an asymmetrical shape, which may suggest the heterogeneity of the reduced phase.

## 4. Conclusions

In summary, magnesium oxide is incorporated into the grain of the wustite catalyst, which was confirmed by both the selective digestion analysis and the XRPD technique. The presence of magnesium oxide in the wustite structure slows its reduction process. After the reduction process, magnesium oxide is on the catalyst’s surface, as was confirmed by the presence of the Auger peaks.

Magnesium oxide acts as a structure-forming promoter in the wustite catalyst, stabilizing the α-Fe crystallites formed during the reduction process. It also acts as an activating promoter influencing its activity. There is a relationship between the activity and the content of magnesium oxide in the catalyst. The content of magnesium oxide above 1.2% in the catalyst grain causes a significant increase in the average size of iron crystallites, which has a negative impact on the activity of the catalysts.

## Figures and Tables

**Figure 1 materials-15-08309-f001:**
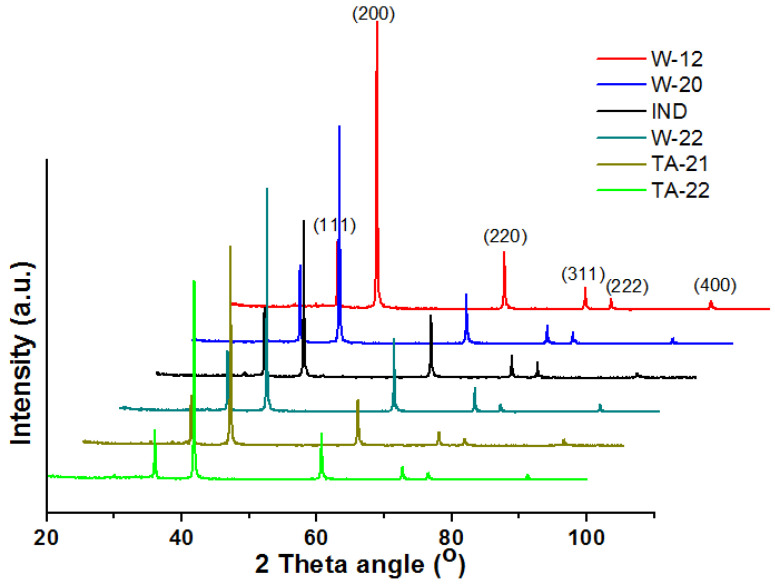
XRPD patterns of catalyst precursors.

**Figure 2 materials-15-08309-f002:**
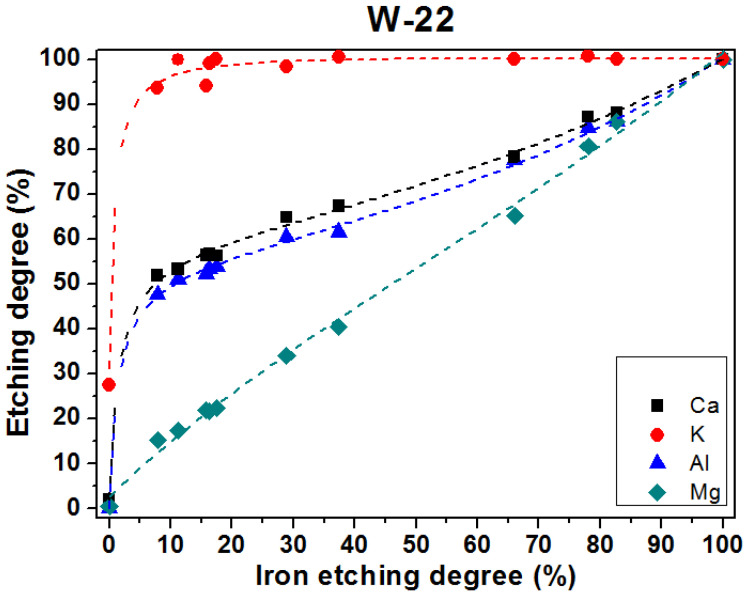
The degree of promotor etching related to the degree of iron etching in the precursor W-22.

**Figure 3 materials-15-08309-f003:**
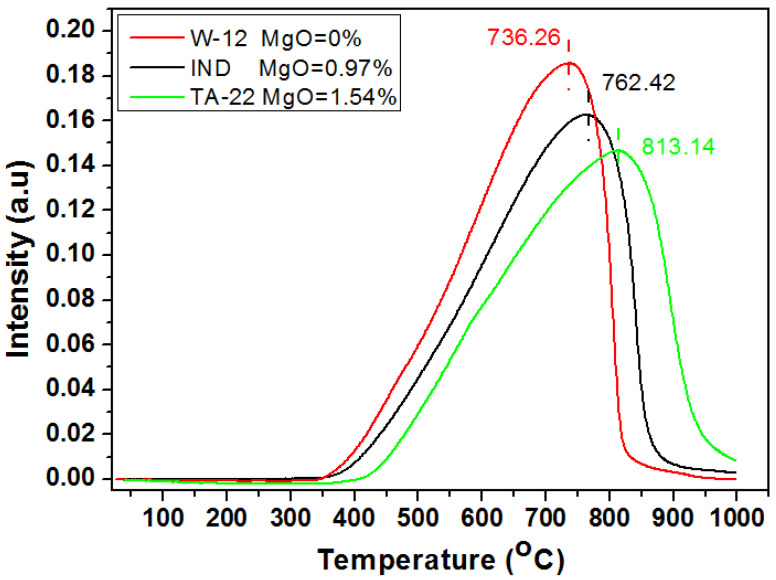
TPR profile of precursors.

**Figure 4 materials-15-08309-f004:**
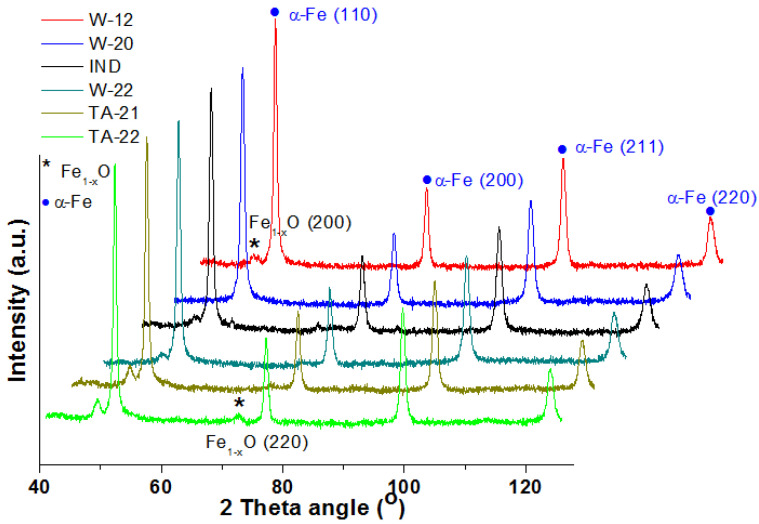
XRPD patterns of the catalysts obtained after reduction of precursors at 500 °C.

**Figure 5 materials-15-08309-f005:**
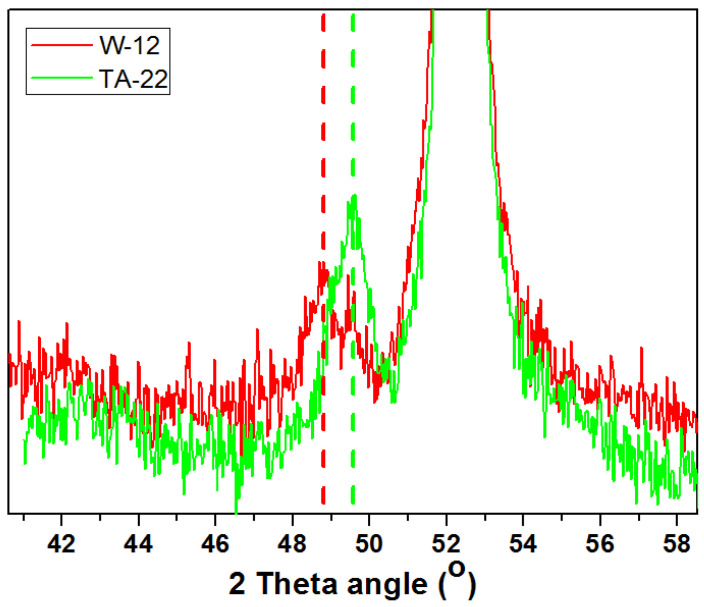
XRPD patterns of the catalysts after reduction at 500 °C, which represent a shift.

**Figure 6 materials-15-08309-f006:**
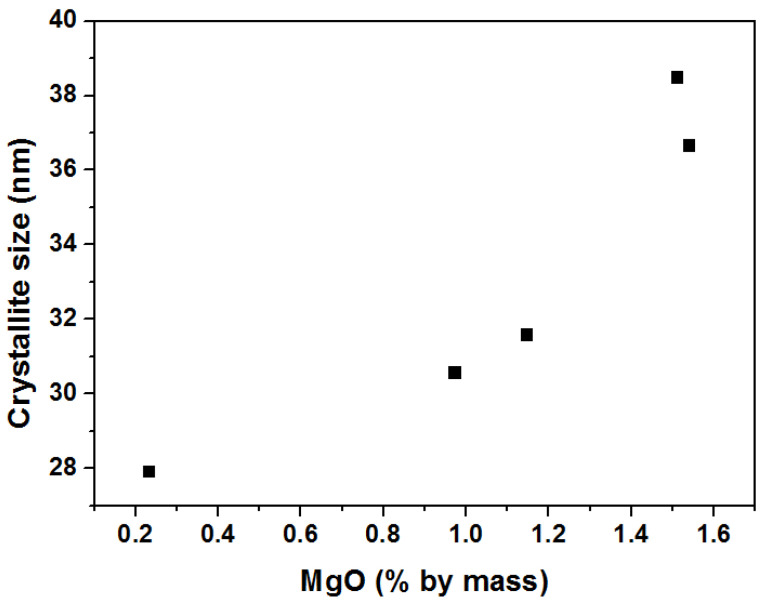
The size of iron crystallites related to the concentration of MgO in the precursor after reduction at 500 °C. Diffraction data were acquired at 25 °C.

**Figure 7 materials-15-08309-f007:**
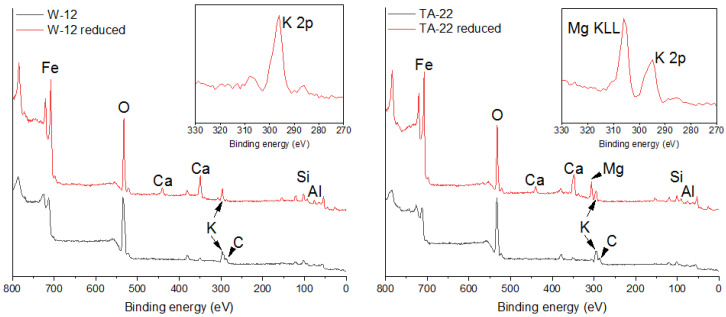
X-ray photoelectron survey spectra of samples W-12 (**left** panel) and TA-22 (**right** panel). The surface compositions before and after reduction in a hydrogen atmosphere are shown. The inserts in both panels represent blow-ups of the binding energy range taken from each survey spectrum after the reduction in hydrogen characteristic of the presence of potassium K 2p photoelectron peaks as well as magnesium Mg KLL Auger electron peaks.

**Figure 8 materials-15-08309-f008:**
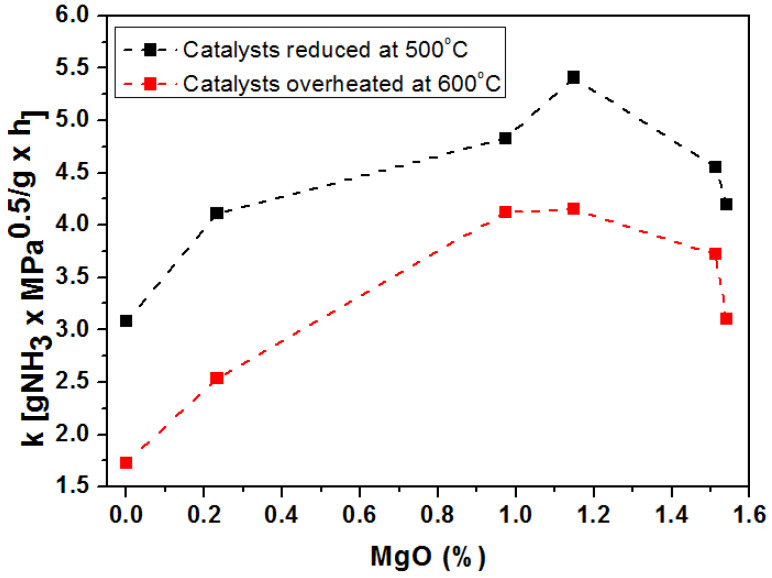
Dependence of the activity of the catalysts (k) measured at 450 °C on the amount of magnesium oxide in the precursors.

**Table 1 materials-15-08309-t001:** Chemical composition and molar ratio Fe^2+^/Fe^3+^ (R) of catalyst precursors.

Sample Name	R	Oxide Content [%]			
		Al_2_O_3_	CaO	K_2_O	MgO
W-12	5.21	2.25	1.74	0.43	0.00
W-20	8.80	2.01	2.02	0.45	0.23
IND	5.34	2.25	2.04	0.49	0.97
W-22	9.46	1.80	1.75	0.37	1.15
TA-21	8.36	2.55	2.16	0.46	1.54
TA-22	6.39	2.72	2.11	0.47	1.51

**Table 2 materials-15-08309-t002:** Concentration of precursors in iron grains.

Sample Name	R	Concentration in Iron Grain [%]			
		Al_2_O_3_	CaO	MgO	K_2_O
W-12	5.21	1.05	0.57	0.00	0.02
W-20	8.80	0.46	0.81	0.13	0.03
IND	5.34	0.89	0.52	0.74	0.04
W-22	9.46	0.99	0.90	1.04	0.03
TA-21	8.36	0.55	0.67	1.22	0.04
TA-22	6.39	0.94	0.85	1.33	0.04

**Table 3 materials-15-08309-t003:** Reduction degree of precursors obtained in the reduction process performed up to 1000 °C with heating rate of 10°/min.

Sample Name	Reduction Degree (%)
W-12	97.8
W-20	95.2
IND	94.2
W-22	89.5
TA-21	86.5
TA-22	86.1

**Table 4 materials-15-08309-t004:** The atomic concentrations of promoters related to the atomic concentration of potassium as observed by XPS.

Sample Name	n_X_/n_K_			
	Silicon	Aluminum	Calcium	Magnesium
W-12	0.3	0.7	0.2	0
W-12 reduced	1.3	1.4	1.3	0
TA-22	0.3	0.7	0.2	0
TA-22 reduced	1.3	1.6	2.0	0.2 (estd.)

## Data Availability

Not applicable.

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
