# Peer review of "Influence of Magnesium Oxide on the Structure and Catalytic Activity of the Wustite Catalyst for Ammonia Synthesis"

_materials, 2022, doi:10.3390/ma15238309_

Round 1

Reviewer 1 Report

This work is devoted to the study of the influence of MgO on the wustite catalysts of ammonia synthesis. Ammonia synthesis is an important product of industrial chemistry and any improvements of this process are actually. I assume that after small corrections this work should be published.

My remarks:

1) Do you have BET measurements for these catalysts? What is the influence of MgO on the summary surface? Please, add these data to manuscript.

2) What is the difference between bulk and surface contents of promotors? XPS results are very brief and it needs to increase. Please, add XPS data of surface concentration elements for comparing with bulk concentrations.

3) What is the reason for increasing catalytic activity with adding MgO up to 1.2%? Do you have any hypothesis?

Author Response

1) Do you have BET measurements for these catalysts? What is the influence of MgO on the summary surface? Please, add these data to manuscript.

The specific surface areas of the reduced catalysts were investigated by the method of single-point nitrogen adsorption at liquid nitrogen temperature. No obvious influence of MgO content on the specific surface area of the catalysts was observed.

2) What is the difference between bulk and surface contents of promotors? XPS results are very brief, and it needs to increase. Please, add XPS data of surface concentration elements for comparing with bulk concentrations.

Estimated atomic concentration of promoters observed on the surface of both catalysts before and after the reduction process was introduced in the manuscript.

3) What is the reason for increasing catalytic activity with adding MgO up to 1.2%? Do you have any hypothesis?

At this moment we don’t have any hypothesis.

Reviewer 2 Report

The manuscript reported the effects of MgO on the structure and catalytic activity of the wustite catalyst for ammonia synthesis.

The author tried to tell the readers the effects of MgO, however, only the change of the FeO structure was shown in the manuscript, and nothing about the catalytic activity could be found. As a catalyst for NH3 synthesis, the long-term stability of the catalysts was the most important issue, which also could not be found. Even some of the results were interesting, however, some important data was missing, which could not be accepted for publication in Materials.

Author Response

Sorry, but I cannot agree with the Reviewer. In the manuscript it has been shown how MgO influences on the activity and thermal stability of the catalysts in the ammonia synthesis.

Figure 8 shows a correlation of MgO concentration in precursors with the activity of catalysts examined at 450°C under 10 MPa. Two series of catalysts were examined: data presented as a black point display activity for catalysts reduced at 500°C, while the red points represent the activity of catalysts after reduction at 600oC. The reduction at 600oC was used to simulate overheating of catalysts in the reactor.

Reviewer 3 Report

This manuscript reported the influence of magnesium oxide admixture on the activation process and the catalytic activity of the ammonia synthesis reaction. A series of characterization methods were performed including XRPD, XPS, TPD etc. However, discourse logic was chaotic and the experiment part was less convincing. The author may consider the following comments and provide a revised version that addresses these points:

1.   In the introduction part, the author mentioned that “The synthesis of ammonia still requires high temperatures and high pressures”. In this text, does the addition of MgO optimize the reaction conditions? Besides, the author mentioned that The role of Al2O3 in the magnetite precursor of the ammonia synthesis catalyst is well known. However, the influence of Al2Oon catalytic performance was not clear in your text. Relevant experimental evidence needs to be presented.

2.   The author listed a variety of samples (W-12, W-20, TN-21). The abbreviation names of the samples need to be explained. What do they represent respectively?

3.   In figure 1 and figure 4, the range of degrees on the x-axis needs to be extended to describe the whole image of W-12 sample. Besides, the positions of the diffraction peaks were obviously shifted, what do they stand for? Or, the author wanted to present the XRPD patterns of different samples in this way. The XRPD patterns in this paper (Energy Environ. Sci., 2020, 13, 5068-5079) would be better.

4.   The authors should supplement the relevant characterization of the catalysts, such as SEM and TEM. As the following valuable paper (CCS Chemistry, 2020, 2, 583-604) showed, the morphology of the sample can be characterized.

5.   In the Supporting Information, all figures lack captions, please add them.

6.   The latest research should be cited and discussed. It is suggested that the authors add a summary and comparison of the catalytic performance for ammonia synthesis of the other samples published in previous papers or commercial products (such as Transactions of Tianjin University, 2020, 26, 6791; CCS Chemistry, 2022, 4, 17581769).

7.   The 20th reference should be in the same format as the other references. And the abbreviation should be written correctly, such as XRPD.

Author Response

  1. In the introduction part, the author mentioned that “The synthesis of ammonia still requires high temperatures and high pressures”. In this text, does the addition of MgO optimize the reaction conditions? Besides, the author mentioned that “The role of Al2O3in the magnetite precursor of the ammonia synthesis catalyst is well known”. However, the influence of Al2Oon catalytic performance was not clear in your text. Relevant experimental evidence needs to be presented.

The introduction briefly presents the current state of research on the process of ammonia synthesis according to the Haber-Bosh technology. Information on the search for new catalysts for this process is presented. The role of Al2O3, which is widely described in the literature, has not been investigated in this paper. At work, it was investigated how the amount of MgO impacts on the reduction process and activity of wustite catalyst in the ammonia synthesis reaction.

  1. The author listed a variety of samples (W-12, W-20, TN-21…). The abbreviation names of the samples need to be explained. What do they represent respectively?

The industrial wustite catalyst is denoted as IND. Samples denoted as W-12, W-20, W-22 TA-21 and TA-22  represent wustite catalysts and number of synthesis we made. These are own names of samples.

  1. In figure 1 and figure 4, the range of degrees on the x-axis needs to be extended to describe the whole image of W-12 sample. Besides, the positions of the diffraction peaks were obviously shifted, what do they stand for? Or, the author wanted to present the XRPD patterns of different samples in this way. The XRPD patterns in this paper (Energy Environ. Sci., 2020, 13, 5068-5079) would be better.

The x-axes in Figures 1 and 4 cover the entire range for all catalysts tested. It is a form of spatial visualization. In Figure 1, the reflex shift was not observed. On the other hand, in Figure 4 we observed a shift of one reflex, which was described in the manuscript.

  1. The authors should supplement the relevant characterization of the catalysts, such as SEM and TEM. As the following valuable paper (CCS Chemistry, 2020, 2, 583-604) showed, the morphology of the sample can be characterized.

The morphology of the iron catalysts is identical for all iron catalysts. Theuse of SEM techniques would not provide useful information when studying bulk catalysts. Such photos we published in our previous (Appl. Catal. A Gen., 247( 2003), 9-15.     https://doi.org/10.1016/S0926-860X(03)00084-X, Applied Catalysis A: General 400 (2011) 48–53, Polish Journal of Chemical Technology, 21, 3, 48—52, 10.2478/pjct-2019-0029).

 TEM technique was not used.

  1. In the Supporting Information, all figures lack captions, please add them.

It was corrected.

  1. The latest research should be cited and discussed. It is suggested that the authors add a summary and comparison of the catalytic performance for ammonia synthesis of the other samples published in previous papers or commercial products (such as Transactions of Tianjin University, 2020, 26, 67–91; CCS Chemistry, 2022, 4, 1758–1769).

In the research, we used the industrial wustite catalyst (IND), which was our reference. On this basis, the positive effect of MgO on the activity and thermostability of the wustite catalyst was determined. The proposed literature has been included in the manuscript.

  1. The 20th reference should be in the same format as the other references. And the abbreviation should be written correctly, such as XRPD.

It was corrected.

Round 2

Reviewer 2 Report

The authors' response was unsatisfactory.

As a catalyst with an industrial background, its long-term use stability must be provided, therefore, the catalyst needs used in the authors' industrial conditions at least 24h, even at several days for a stability experiment; however, the author did not provide any corresponding data, therefore, I persevered to my original opinion.

Author Response

Thank you for your revision of my manuscript. I agree that data on overheating conditions was missing in the manuscript. 

Thermostability was evaluated by overheating the catalyst under the atmosphere of the synthesis gas at pressure of 0.1 MPa over period 17 h at temperature 600 oC. After overheating was completed, the determination of the catalyst activities was performed again.

In the experimental part, information on the conditions of thermostability test has been added.

Reviewer 3 Report

After reading the reply letter, the authors did a part revision on the draft,  I think the author should pay more attention to the logic of the article and modify the corresponding content especially for the first and third paragraph related to the ammonia synthesis using Al2O3  etc.

Author Response

Thank you for your revision of my manuscript. The introduction was improved.